# Bacteriophages and Endolysins Used in the Biocontrol of *Staphylococcus aureus*

**DOI:** 10.3390/microorganisms13112638

**Published:** 2025-11-20

**Authors:** Maryoris E. Soto Lopez, Ana Margarita Otero-Herrera, Fernando Mendoza-Corvis, Jose Jorge Salgado-Behaine, Rocio López-Vergara, Ana M. Hernández-Arteaga, Derrick Cortessi, Pedro M. P. Vidigal, Omar Pérez-Sierra

**Affiliations:** 1Research Group on Food Properties and Processes—GIPPAL, Department of Food Engineering, Faculty of Engineering, University of Córdoba, 6th Avenue N° 76-103, Monteria 230002, Colombia; anaoteroh@correo.unicordoba.edu.co (A.M.O.-H.); fmendoza@correo.unicordoba.edu.co (F.M.-C.); josesalgadob@correo.unicordoba.edu.co (J.J.S.-B.); rociolopezv@correo.unicordoba.edu.co (R.L.-V.); anahernandeza@correo.unicordoba.edu.co (A.M.H.-A.); oaperez@correo.unicordoba.edu.co (O.P.-S.); 2Animal and Dairy Sciences Department, University of Wisconsin-Madison (UW-Madison), Madison, WI 53706-1205, USA; dabold@wisc.edu (D.C.); pedro.vidigal@ufv.br (P.M.P.V.); 3Núcleo de Análise de Biomoléculas (NuBioMol), Campus da UFV, Universidade Federal de Viçosa (UFV), Viçosa 36570-900, MG, Brazil

**Keywords:** antimicrobial strategies, bacteriophages, endolysins, food safety, food industry, methicillin-resistant *Staphylococcus aureus* (MRSA)

## Abstract

*Staphylococcus aureus* is a major foodborne pathogen associated with contamination of dairy and meat products, posing a persistent challenge to food safety due to its biofilm formation and resistance to multiple antibiotics. In this review, we summarize recent advances in the use of bacteriophages and phage-derived endolysins as targeted biocontrol agents against *S. aureus* in food systems. Bacteriophages exhibit host specificity and self-replicating capacity, while endolysins provide rapid lytic activity, minimal resistance development, and effectiveness against biofilm-embedded cells. Studies demonstrate significant microbial reductions in milk, cheese, and meat matrices, although factors such as pH, salt, and fat content can influence their efficacy. The integration of these biocontrol tools into food preservation represents a sustainable and safe alternative to conventional antimicrobials. Finally, we discuss current limitations and the need for optimizing formulations, stability, and regulatory frameworks to facilitate the adoption of phage and endolysin-based products in the food industry.

## 1. Introduction

*Staphylococcus aureus* is a Gram-positive anaerobic bacterium that groups in clusters and is β-hemolytic, catalase and coagulase-positive. It is widely distributed in various environments and present in the microbial flora of humans, which increases the risk of its dissemination [1]. It is one of the etiological agents frequently found in clinical settings and food processing plants. Furthermore, it produces various enterotoxins that are heat stable, capable of forming biofilms and evolving into strains resistant to multiple drugs such as antibiotics, such as methicillin-resistant *S. aureus* strains (MRSA) and Vancomycin-Resistant *S. aureus* (VRSA) [2].

MRSA and VRSA exhibit a high mutation rate and adaptation related to a short duplication period in the environment in which they are found. MRSA causes ten times more infections than all multi-resistant Gram-negative pathogens combined [3]. According to the report published by the European Food Safety Authority and the European Centre for Disease Prevention and Control, *S. aureus* is one of the main pathogens that caused hospitalizations due to foodborne outbreaks in 36 European countries in 2018 [4]. Therefore, prevention of food contamination by *S. aureus* has generated growing interest in different areas, such as the food industry [5].

Among the traditional treatments used to control *S. aureus* are antibiotics, resins, essential oils, and peptides with antimicrobial properties [6]. Currently, one of the proposed alternatives to deal with the emergence of multi-resistant strains is the use of bacteriophages. Bacteriophages are natural antimicrobials that exist widely in the environment. They can overcome the barriers generated by bacterial defense mechanisms and have a high specificity to bind to their host through receptor-binding proteins (RBPs) that recognize specific molecules present on the surface of bacteria. The high specificity of bacteriophages allows them to attack pathogenic bacteria without affecting the microbiota [7,8,9].

Bacteriophages can be lytic or lysogenic, with the difference being found in the cycle they use to infect the bacterial host. In the lytic cycle, holins perforate the cytoplasmic membrane of bacteria, thus generating access to the peptidoglycan so that endolysins can degrade the bacterial cell wall, lyse the host, and release new bacteriophages into the external environment [10]. In the lysogenic cycle, viral genetic material is incorporated into the bacterial chromosome, with which it replicates and then is transferred to new bacteria without lysis of the host. During the phage-host interaction, virulent genes can be expressed, and the lysogenic life cycle can be converted to a lytic cycle [11,12]. In bacteriophage treatment to control bacterial contamination, the use of lysogenic bacteriophages is avoided due to the possibility of horizontal transfer of resistance or virulence genes to other bacteria lacking these genes [13].

Most *S. aureus* bacteriophages are lysogenic. They have been identified in the DNA of the microorganism, contributing to bacterial resistance due to the encoding of virulence factors. However, it is important to mention that exclusively lytic phages can also be found [14]. Bacteriophage JD419 for the biocontrol of *S. aureus* strains N315, MR-84, and MS-41, which was found to be a lysogenic phage by genomic analysis, had a latency period of 50 min and burst size of 33 PFU/infected cell; therefore, to improve its bactericidal activity and avoid genes related to the lysogenic cycle, it is necessary to make modifications through genetic engineering [15].

On the other hand, endolysins produced by bacteriophages that infect *S. aureus* generally have catalytic domains, such as an N-terminal, cysteine, histidine-dependent amidohydrolase/peptidase domain and a central N-acetylmuramoyl-l-alanine domain [16]. Endolysins have been used for the biocontrol of *S. aureus* in vitro and in foods contaminated with the microorganism, mainly in meat and dairy products. The endolysin LysSAP27 obtained from the phage vB_SauS-SAP27 (ϕSAP27), which had previously shown little efficiency in inhibiting the growth of *S. aureus*, was used in milk samples contaminated with *S. aureus* ATCC 6538, where biological control could be determined during treatment [17].

The use of bacteriophages and phage-derived endolysins represents a potential alternative for the biocontrol of *S. aureus* strains in food, which cause food spoilage and foodborne illnesses. Previous reviews on *S. aureus* bacteriophages have reported their genomic and morphological characteristics and their efficiency in the treatment of skin infections [18,19]. Therefore, this review focuses on the applications of bacteriophages and endolysins for the biocontrol of *S. aureus* in food, the characteristics of the microorganism’s bacteriophages, and the importance of endolysins in the food industry. Additionally, it covers commercial phage products for the biocontrol of *S. aureus* and the main limitations of *S. aureus* bacteriophages and endolysins in their application in food products.

## 2. *S. aureus* in the Food Industry

*S. aureus* is a pathogen that can be transmitted through the consumption of food contaminated during preparation or processing, either by direct contact or by respiratory secretions from food workers. It grows at temperatures between 7 and 48.5 °C, pH 4.2 and 9.3, and a_w_ as low as 0.83; it grows optimally near 30–37 °C, pH 7.0–7.5, and a_w_ 0.99, conditions frequently encountered in many food matrices [20,21,22].

Specific processing conditions (temperature, salt, and nutrient availability) can promote *S. aureus* growth and toxin expression (leukocidins, hemolysins, and staphylococcal enterotoxins, classical: SEA-SEE, non-classical: SEG, SHE; and enterotoxin-like: SElW, SElX, etc.), which underlie host survival and staphylococcal food poisoning with emesis [22,23,24]. These enterotoxins are pyrogenic and highly resistant to acidic pH, heat, and proteolysis, so the risk may persist even after certain treatments [25,26].

*S. aureus* has the ability to adhere, colonize, and form biofilms throughout the food production chain. This bacterium exhibits a strong adhesion capacity to various surfaces, particularly stainless steel and polystyrene. The extensive use of polystyrene in food processing environments increases the risk of recurrent cross-contamination [27,28]. Biofilm development in *S. aureus* is largely regulated by the intracellular adhesion (ica) locus, which encodes enzymes involved in the synthesis of the polysaccharide intercellular adhesin (PIA). The icaA and icaD genes play a central role in the process, catalyzing the polymerization of N-acetylglucosamine, the main exopolysaccharide that mediates intracellular adhesion and biofilm accumulation. The expression of the ica operon is further modulated by global regulators such as sarA, which enhances biofilm formation, and agr, which typically represses it in favor of acute virulence. Nevertheless, biofilm mass can vary even among strains harboring the ica locus, suggesting the involvement of ica-independent mechanisms, including surface proteins such as protein A and fibronectin-binding proteins [29]. Strains of *S. aureus* isolated from foods, processing facilities, and food handlers have demonstrated the ability to form biofilms on both stainless steel and polystyrene surfaces [30].

Infections and staphylococcal food poisoning linked to dairy and meat products contaminated with *S. aureus* have been reported, often associated with poor hygiene during milking, handling, transportation, and storage. These foods can serve as vehicles for transmission from farm to consumer; therefore, preventing and controlling *S. aureus* outbreaks along the food chain is essential [31,32]. Figure 1 shows the transmission route and the key problem points associated with *S. aureus* contamination in the industry.

## 3. Characteristics of *S. aureus* Bacteriophages

*Staphylococcus* bacteriophages have an icosahedral capsid and a tail, carry linear double-stranded DNA, and belong to the class Caudoviricetes according to the recent update of the International Committee on Taxonomy of Viruses (ICTV). Electron microscopy reveals tail morphologies that correlated with typical genome sizes: 16–18 kb with short tails, 39–43 kb with long non-contractile tails, and 120–140 kb with long contractile tails, often with double sheath [19,33].

Prophages in *S. aureus* have been reported with a 39–43 kb genome and a long non-contractile tail. Their genomes display a modular organization, including lysogeny, DNA replication, transcription regulation, packaging, and head, tail, and lysis modules [34]. The lysogeny module encodes integrase (*int*) genes that recognize *attP* (phage) and *attB* (bacteria) sites and mediate unidirectional site-specific integration of the phage genome into the bacterial chromosome [35]. Prophage carriage can modulate host fitness, pathogenicity, and survival [36].

Among the lysogenic bacteriophages of *S. aureus* most widely studied are ϕ11 and ϕ80α, which show high transduction efficiency and are used for genetic engineering between *S. aureus* strains [37]. Phages in the same group have also been reported to carry lysogeny genes within the DNA regulation/manipulation module but lack the *cI* gene; consequently, they do not establish lysogeny and do not mediate horizontal gene transfer to *S. aureus* strains. The SA97 phage encodes an integrase (SA97_040); however, no virulence or drug-resistance genes were detected, supporting its use as a biocontrol agent against *S. aureus* [38].

On the other hand, bacteriophages belonging to group A represent a highly conserved lineage in terms of nucleotide and amino acid homology, lytic infection mechanism, genome size, GC content (27–29%), and predicted gene number (20–29%). Approximately 60% of their genes are associated with essential functions such as DNA replication (DNA-binding proteins, DNA polymerase), virion morphogenesis (DNA packaging, tail fibers, collar, and major capsid proteins), and host cell lysis (holin and endolysin) [36]. These phages exhibit remarkable genome compaction (<20 kb) and a streamlined genetic organization primarily dedicated to replication and structural functions, leaving minimal space for accessory or virulence-related genes. Their compact and well-defined genomes, together with a strictly lytic lifestyle, restrict uncharacterized regions and mobile genetic elements, thereby reducing the potential for horizontal gene transfer and enhancing their biosafety for therapeutic and biocontrol applications [39]. Among the most extensively studied members of this group are phages S13′ and S24-1, which have been effectively used against infections caused by *S. aureus* strains [40].

Long-tailed contractile staphylococcal phages encode 200–250 open reading frames and several tRNA genes, showing high nucleotide similarity (88.3–99.9% identity). They carry linear dsDNA of approximately 127–145 kbp with direct terminal repeats; notable examples include Stau2, StAP1, vB_SauM_Remus, vB_SauM_Romulus, SA11, and qdsa001, with high average nucleotide identity values (>95%) and shared gene content (>77%) [36]. Their genomes display a modular organization encompassing morphogenesis, cell lysis, nucleic-acid metabolism, and host metabolic reprogramming [41]. Infections typically show average latency periods of 30–50 min [42]. Table 1 below shows *S. aureus* bacteriophages and their main characteristics.

## 4. Biocontrol of *S. aureus* with Bacteriophages

In recent years, various investigations have been carried out where the biocontrol of *S. aureus* has been evaluated, especially in antibiotic-resistant strains. Specifically using lytic bacteriophages classified in groups A and C, which exhibit broader host ranges than those with a lysogenic nature (group B) [49]. Among the highlighted factors that have been evaluated for the biocontrol of *S. aureus* are the characterization of bacteriophages, the optimization of temperature conditions, and the time in which maximum reductions in the bacteria are generated, the MOI of the bacteriophages used to lyse *S. aureus* strains.

The vB_SauS_IMEP5 phage showed a 30 min latency and burst size of 272 PFU/infected cell; its lytic activity against *S. aureus* (6.6 × 10^8^ CFU/mL) was tested across MOIs 10^−4^–100, with MOI 10^−3^ as the optimum; the authors reported high bacteriolytic potential and good efficiency in limiting bacterial growth [50]. The vB_SauS_SA2 phage exhibited a 10 min latency and a burst size of 293 PFU/infected cell; its bactericidal effect was assessed in *S. aureus* F2 (5.5 × 10^8^ CFU/mL) at MOIs 10^−5^ and 1 for 24 h at 3 °C. At MOI 1, lysis began 2 h after co-incubation, and by the end of treatment, most of the bacteria were lysed at all MOIs, indicating a strong inhibitory effect and high replication rate in *S. aureus* F2 [51].

The lytic activity of two lytic bacteriophages, vB_SauM-515A1 and vB_SauP-436A, of the commercial *Staphylococcus* bacteriophage cocktail produced by Microgen (Russia) was evaluated at MOI 0.1 and 1. The bactericidal effect of bacteriophage vB_SauM-515A1 was evaluated on *S. aureus* SA515 cultures, while the inhibitory effect of bacteriophage vB_SauP-436A1 was determined on *S. aureus* SA436. The results indicated that lysis of bacterial culture of the host strain *S. aureus* SA515 was faster than that of the *S. aureus* SA436 strain for approximately 3 h, which is attributed to the shorter latency period and larger burst size of bacteriophage vB_SauM-515A1 (40 min and 185 PFU/infected cell) compared to bacteriophage vB_SauP-436A1 (50 min and 94 PFU/infected cell) [52].

The UPMK_1 phage (MOI 1) exhibited a 20 min latency and a burst size of 32 PFU/infected cell. In *S. aureus* MRSA t127/4, treatment reduced optical density versus the untreated control, with a marked effect in the first 4 h and persistence up to 8 h [43]. In a separate assay, CSA13 (MOI 1) against *S. aureus* (2.6 × 10^8^ CFU/mL) decreased optical density to undetectable from the first hour, remaining so up to 20 h; CSA13 showed a 20 min latency and a burst size of 230 PFU/infected cell [33]. Collectively, these studies confirm the promising potential of lytic bacteriophages as biocontrol tools against *S. aureus*, including antibiotic-resistant strains.

Beyond lytic efficiency, the stability of *S. aureus* bacteriophage under varying physicochemical and environmental conditions is a key determinant of their applicability in food systems. Phages generally remain viable between 4 °C and 50 °C and within a pH range of 4.0–11.0, but they are rapidly inactivated at temperatures ≥ 70 °C or under extreme acidic (pH ≤ 3.0) environments [53]. Complex foods denature phage particles, diminishing infectivity [54]. To enhance stability, preservation strategies such as lyophilization with cryoprotectants (e.g., trehalose, sucrose) or microencapsulation within polymeric matrices (e.g., alginate, chitosan) have been successfully applied to maintain structural integrity and lytic functionality under adverse conditions [55].

Importantly, the environmental impact of phage-based interventions is considered minimal, as bacteriophages specifically infect bacterial hosts and do not affect eukaryotic cells. Their high specificity and natural biodegradability position them as sustainable alternatives to chemical disinfectants or antibiotics, minimizing non-target effects and ecological disturbances. To ensure biosafety, only strictly lytic phages should be employed, thereby avoiding risks of lysogeny and horizontal gene transfer [56]. Supporting this, in situ assessments demonstrated that phages such as FoX2 and FoX4 did not alter local biomass or microbial diversity in soil microbiomes, confirming their environmental compatibility and alignment with sustainable biotechnological [57].

The results obtained in these studies confirm the potential of using bacteriophages for the biocontrol of *S. aureus* in strains resistant to antibiotics. Recently, several investigations have been published related to the isolation, evaluation of the lytic activity, and the efficiency of phages to lyse *S. aureus* present in foods.

## 5. Application of Bacteriophages That Infect *S. aureus* in Foods

In the food industry, bacteriophages and bacteriophage cocktails have been used for biocontrol of *S. aureus* strains present in dairy and meat products in order to decrease the likelihood of bacterial infection and disease, prevent biofilm formation, preserve the product, and promote safe environments in the production, processing, and handling of these products [58].

### 5.1. Milk Products

Milk is one of the most consumed foods worldwide. Through its processing, dairy products such as yogurt, cheese, whey, and butter can be obtained. During processing, thermal treatments are used that reduce the bacterial load and prolong the shelf life of the food; however, in some cases, the presence of microorganisms in these products has been determined [59] because milk and dairy products are often nutrient-rich and pH neutral, allowing the growth of various bacteria such as *S. aureus* [60]. In a study, the prevalence of *S. aureus* in samples of commercial pasteurized milk was determined, indicating that 75% of the isolates were resistant to multiple antibiotics at the different concentrations tested and were capable of producing biofilms [61].

Multiple studies report effective biocontrol of *S. aureus* in whole, skimmed, pasteurized, and UHT milk using bacteriophages, aided by the liquid matrix that facilitates phage-bacterium interaction [62]. In UHT milk inoculated at 10^8^ CFU/mL, adding vB_SauM_ME126 and vB_SauM_ME18 at 25 °C for 6 h showed no control at MOI 1 [63], likely because milk proteins coat the bacterial surface and block host receptors [64]. At MOI 10, bactericidal activity was evident: vB_SauM_ME18 fully reduced the bacteria by 4 h, and vB_SauM_ME126 by 6 h.

In pasteurized milk inoculated with *S. aureus* (10^6^ CFU/mL), phages SA2 and SANF applied at MOI 10^2^ achieved a 90–99% growth reduction after 6 h at 25 °C; by contrast, at 37 °C, only SA2 maintained a >65% reduction over the same period [64]. In UHT milk inoculated at 10^4^ CFU/mL, phage M8 at MOI 10^2^ produced a 3 log_10_ CFU/mL decrease after 8 h at 37 °C [65]. These findings are noteworthy because low host densities typically require a higher phage-to-bacterium ratio to ensure efficient infection and sustained biocontrol [54].

The evaluation of the bactericidal effect of phage SA13m in pasteurized whole milk, previously inoculated with *S. aureus* ATCC 29213 (10^5^ CFU/mL), indicated that adding the phage at MOI 10^2^ achieved complete lysis at 4 °C, lowering counts by 4.1 log_10_ CFU/mL within 24 h with no regrowth up to 48 h. At 25 °C, the reduction reached 4.0 log_10_ CFU/mL by 48 h. Increasing the dose to MOI 10^3^ at 25 °C produced the fastest and largest decline (4.2 log_10_ CFU/mL in 3 h) compared with 4 °C (4.2 log_10_ CFU/mL in 12 h) [66]. In skimmed milk containing *S. aureus* BM001 (10^3^ CFU/mL), addition of bacteriophage LSA2308 under the same temperature–time conditions showed no evident control at MOI of 10^2^–10^3^ at 4 °C; by contrast, at 25 °C, biological control was observed between 6 and 48 h. The best results occurred with an MOI of 10^4^ at 4 °C, where *S. aureus* BM001 was gradually reduced by 99.95% at 48 h [9].

The addition of bacteriophage SA46-CTH2 with MOI 10^4^ in pasteurized milk containing *S. aureus* SA46 (10 ^5^ CFU/mL) at 4 °C for 24 h, reduced the viable counts of the microorganism by 2.2 log_10_ CFU/mL. An increased lytic activity was also reported for a bacteriophage cocktail composed of STA1.ST29, EB1.ST11 and EB1.ST27, applied at an MOI of 10^4^ against *S. aureus* 10614 (1.1 × 10^5^ CFU/mL) in pasteurized milk at 4 °C for 8 h, resulting in a reduction of approximately 2 log_10_ CFU/mL [67]. Similarly, when evaluated in raw milk under the same conditions, the maximum reduction achieved was around 1 log_10_ CFU/mL [68]. The use of bacteriophage cocktails offers an effective alternative for controlling foodborne pathogens such as *S. aureus.* Their combined action broadens the host range and decreases the likelihood of bacterial resistance. Optimizing their formulation involves selecting complementary phages that maximize bacterial reduction while maintaining manageable complexity. This strategy enhances robustness and ensures sustained lytic efficacy across diverse strains and food matrices [69]

### 5.2. Cheese

*S. aureus* has been identified in dairy products and high-salt foods such as cheese [70]. The biocontrol of *S. aureus* present in cheeses has been studied mainly in cheddar cheese during its ripening and in the commercial product. One study evaluated the addition of a cocktail of three bacteriophages, phi812, 44AHJD, and phi2, of different morphologies (belonging to Groups A, B, and C) with MOIs of 15, 45, and 150 in *S. aureus* SMQ-1320 (10^6^ CFU/mL) during the elaboration of cheddar cheese. This reduced the contamination to 4 log_10_ CFU/mL after a ripening period of 14 days at 4 °C, transforming the contaminated milk samples into pasteurized cheese with controlled ripening [71]. On the other hand, the lytic effect of bacteriophage KMSP1 was evaluated with MOI 10^3^ and 10^4^ on *S. aureus* (10^4^ CFU/cm^2^) present in 3 × 3 cm^2^ slices of cheddar cheese, where reductions between 0.6 and 1.4 log_10_ CFU/mL were presented after 30 min at 25 °C; this result was maintained without bacterial multiplication for 24 h [72].

In fresh cheese, gelatin-glycerol films loaded with phiIPLA-RODI (1.75 × 10^8^, 1.16 × 10^8^ and 6.35 × 10^7^ PFU/mL) were tested against *S. aureus* (10^5^ CFU/mL). The highest phage dose yielded the lowest counts (44 CFU/g), whereas the lowest dose sustained control up to day 3 [73]. By day 6, bacterial levels were similar to the control, likely because phages adsorb to the food surface and, given their non-motile phenotype, limit contact with cells [74]. Accordingly, higher phage concentrations and full surface coverage are advised to maximize phage-host contact [75].

The phage cocktail (vB_SauS-phi-IPLA35 and vB_SauS-phi-IPLA88) effectively reduced *S. aureus* in cheese. In fresh cheese, counts decreased by 3.83 log_10_ CFU/g within 3 h, reaching undetectable levels at 6 h. In hard cheese, reductions reached 4.64 log_10_ CFU/g in the curd and 1.24 log_10_ CFU/g after ripening, demonstrating consistent efficacy during processing and storage [76]. Similarly, the *S. aureus* phage cocktail (EBTH and K2, 10^8^ PFU/mL) efficiently controlled both inoculated and native *S. aureus* in traditional Egyptian cheeses. In soft varieties (Karish and Domiati), bacterial counts dropped below detection (≤10 CFU/g) within 3–24 h, preventing enterotoxin accumulation, whereas in hard Ras cheese, complete elimination occurred after 60–90 days of ripening [77]. More recently, a biopreservation strategy for hard Dutch-type cheese using two virulent phages (No. 4 and No. 8) applied post-pasteurization (10^8^ PFU/mL) completely prevented *S. aureus* growth during 60 days of ripening, while the control reached 154.2 CFU/g, confirming the strong protective capacity of the phage treatment [78].

Collectively, these studies demonstrate that the use of lytic bacteriophage and phage cocktails can effectively reduce or eliminate *S. aureus* in both fresh and hard cheeses without compromising the activity of starter cultures or altering the physicochemical properties of the final product. This highlights their potential as safe and efficient biocontrol agents in the dairy industry.

### 5.3. Meat Products

*S. aureus* has also been identified in beef, lamb, chicken, and meat products, causing staphylococcal food poisoning with vomiting and diarrhea [79]. Moreover, the biocontrol of the microorganism (10^8^ CFU/mL) was evaluated in samples of raw chicken filets using the bacteriophage vB_SauM_CP9 with MOI 1 and 10, which presented lytic activity in ten strains of *S. aureus*, with a burst size of 228 PFU/infected cell and a latent period of 45 min. With MOI 1 and 10, the reductions in *S. aureus* were 73.06% and 76.94%, respectively. In the research, they also evaluated the bactericidal effect of 1% thyme oil (*Thymus vulgaris*) in combination with the bacteriophage with an MOI of 10. A reduction of 87.22% of the microorganism was presented for 120 min, which corresponded to the treatment with the greatest inhibition [80]. This result indicates the potential of combined treatment with bacteriophages and antimicrobial products to eliminate foodborne pathogens.

Biocontrol of *S. aureus* in dairy and meat products with bacteriophages has shown varying levels of success. In this regard, the application of endolysins obtained from bacteriophages has been explored as an alternative for biocontrol of the microorganism to overcome the barriers developed by *S. aureus* against phage activity.

## 6. Importance of Endolysins in the Food Industry

The study of the mechanisms by which bacteriophages lyse bacteria has made it possible to determine the antimicrobial potential of endolysins. Endolysins are phage enzymes synthesized during the final stage of the lytic cycle. These enzymes cause the rupture of the bacterial cell wall, thus releasing the newly formed virions so that they can infect new host cells [81]. Endolysins are enzymes composed of the catalytic domain present in the N-terminal region, which provides the catalytic activity, and the binding domain. This domain specifically recognizes and binds to its target bacterial cell wall receptors, present in the C-terminal region. Structurally, endolysins are classified into modular, bacteriophage-encoded, Gram-positive-targeting, bacteria-targeting, and globular, Gram-negative-targeting, which comprise a single enzymatically active domain and lack a modular structure [82,83].

Endolysins are a promising option for food-industry decontamination; they can be produced and purified at scale at low cost, display high specificity for Gram-positive bacteria (acting externally in the absence of an outer membrane), and can be protein-engineered to enhance bacteriolytic activity, solubility, and other physicochemical traits. They also show a lower propensity for resistance because they cleave highly conserved bonds in the peptidoglycan. Taken together, these advantages support their use as biocontrol tools in foods [84,85,86].

Another advantage of endolysins for food industry decontamination is that they spare the beneficial microbiota owing to their host specificity. They also achieved rapid lysis of Gram-positive bacteria in foods within minutes using low doses [87]. Endolysins are likewise promising against biofilms, surface-attached microbial communities, of concern because they drive food contamination, contribute to the pathogenesis of foodborne diseases, and show tolerance to antibiotics and sanitation. The endolysin LysCSA13, derived from the virulent CSA13 phage of *S. aureus*, demonstrated high efficiency (80–90%) in reducing biofilm biomass on polystyrene, glass, and stainless Steel [88].

Figure 2 presents the main advantages of studying and applying endolysins in the food industry. Therefore, endolysins can be used as biocontrol agents for various uses in the food industry, covering applications in equipment, surfaces, and foods.

## 7. Biocontrol of *S. aureus* with Endolysins

Regarding the studies carried out evaluating the lytic activity of endolysins in *S. aureus* strains, the endolysin LysSAP8 derived from the bacteriophage SAP8 presented high efficiency in the inhibition of *S. aureus* at temperatures between 15 and 30 °C, pH between 7 and 8, in a medium supplemented with calcium ions and a NaCl concentration of 500 mM. The evaluation of the antimicrobial effect on *S. aureus* KCCM 12103 (>10^8^ CFU/mL) by LysSAP8 (1 µM) resulted in a reduction of 3.46 log_10_ (CFU/mL) in 30 min, evidencing that endolysin has a good potential for applications in foods containing NaCl or neutral/subacidic foods [89].

Endolysin HY-133 showed a Minimum Inhibitory Concentration (MIC) of 0.12–0.25 µg/mL against methicillin-resistant *S. aureus* (MRSA), methicillin-susceptible *S. aureus* (MSSA), and mupirocin-resistant *S. aureus* (MupRSA) tested at 5 × 10^5^ CFU/mL. Maximal inhibition occurred at 2 h, comparable to daptomycin (a lipopeptide antibiotic). However, regrowth by 48 h was observed for MSSA ATCC 29213 and MRSA ATCC 43300 [90]. For CF-301 (exebacase), MICs of 0.2–0.4 µg/mL were reported with lysis of *S. aureus* ATCC 29213 (4.8 × 10^5^ CFU/mL) [91]. In another study, recombinant LysSAP26 showed an MIC of 20 µg/mL across 20 oxacillin-resistant *S. aureus* (ORSA) isolates [92].

The lytic activity of the CHAP-amidase domain of endolysin Lysk (1, 5, and 10 µg/mL) was evaluated in MRSA strains (8 × 10^8^ CFU/mL). With 10 µg/mL of Lysk, the greatest reduction in MRSA (3.2 Log_10_ CFU/mL) was observed during 50 min of incubation at 37 °C, while a MIC of 1 µg/mL reduced MRSA by more than 1 log_10_ CFU/mL. Regarding temperature, there was no significant alteration in CHAP-amidase activity between 5 and 60 °C for 15 min. Furthermore, endolysin remained active for 1 month and 1 year when stored at 4 °C and −20 °C, respectively [93].

The evaluation of the lytic activity of the CHAPk endolysin was evaluated in MRSA 252 (7 × 10^5^ CFU/mL) at temperatures of 32 and 37 °C, the latter being the one in which the greatest decrease in the bacterial optical density of MRSA 252 was observed [94]. The evaluation of the bacteriolytic effect of endolysin LysRODI (13.44 μg/mL) derived from the bacteriophage phiIPLA-RODI was determined in 67 MRSA strains. The results obtained showed that endolysin had a lower inhibitory effect in 62.69% of the strains, such as MRSA-IPLA 4, 20, 22, 26, 30, 32, 35, 43, 50, and 58. The medium and high lytic activity was present in the remaining 37.31% of the strains, among which the activity in MRSA-IPLA 28 and 46 stood out, respectively [95].

## 8. Application of Endolysins Infecting *S. aureus* in Foods

The lytic activity of endolysins for the biocontrol of *S. aureus* has been studied mainly in dairy products such as pasteurized and UHT whole milk, skimmed milk, cheese, and in meat products such as pork, beef, ham, and bacon. The lytic activity of the endolysins evaluated has shown variability in the reduction of the microorganism count in foods. The variability observed is mainly due to the protein stability and antimicrobial efficiency of the endolysins used. In turn, this is closely related to various factors such as the structure of the enzyme, the use of cofactors, the specificity, and affinity that endolysins present with foods, related to their composition of fats, proteins, minerals, among others, in addition to biochemical factors such as temperature, pH, and ionic strength that occur when carrying out the treatments [96].

### 8.1. Endolysin-Based Control in Milk Products

Regarding the biocontrol of *S. aureus* in milk using endolysins, the antimicrobial effect of LysGH15 (25 nM) was determined in whole and skimmed milk samples contaminated with MRSA 2701 (10^8^ CFU/mL). This was performed to evaluate the effect of fat on the antimicrobial efficiency of LysGH15 in the milk samples evaluated at 37 °C for 24 h. The results indicated that LysGH15 did not present a significant bactericidal activity, and the growth rate of MRSA increased slightly, similar to the behavior in the control group. The bacteriolytic action occurred more quickly in skimmed milk than in whole milk due to endolysin, signifying that the lower amount of fat was more favorable for the bactericidal activity; this is attributed to the fact that fat affected the binding of LysGH15 to MRSA 2701 cells [97].

Similar results were presented in the study of the antimicrobial activity of endolysin LysSA97 (1.88 μM) in previously pasteurized skimmed and whole milk samples inoculated with *S. aureus* RN4220 (10^5^ CFU/mL). The reduction in bacterial cell count was less than 2 log_10_ CFU/mL in skimmed milk for 3 h, while in whole milk samples, there was no evident control of *S. aureus*. Therefore, endolysin may be more effective in low-fat food products [98].

The lytic activity of Lys109 (30, 300, 900, and 1500 nM) was determined against the MRSA strain CCARM 3090 (10^5^ CFU/mL) previously inoculated in pasteurized whole milk at 37 °C for 1 h. After treatment with 300 nM of Lys109, the control of the bacteria was evident, taking into account that the reduction had a value of 2 log_10_ CFU/mL of *S. aureus* at the end of the evaluation. Regarding the results related to the concentration of 900 nM of lysine, there was a total elimination of the microorganism after 45 min of testing [99].

The inhibitory effect of LysSA11 at concentrations of 1.125, 2.25, 3.375, 4.5, and 9 μM on MRSA CCARM 3089 (2 × 10^5^ CFU/mL) present in commercial pasteurized milk was evaluated for 1 h at 4 °C and 25 °C, respectively. Endolysin showed greater activity at 25 °C, since bacterial cells were reduced to undetectable levels with 9 μM of lysine in 30 min. At the same concentration at 4 °C, the same results were obtained after 60 min of treatment. This indicates an adequate temperature for applying the enzyme [100]. Therefore, it should be used after heat treatments in milk or other foods and should not be applied to refrigerated products [101].

Comparison of the potential of LysRODI lysine (4 μM, 8 μM, or 16 μM) obtained from phage phiIPLA-RODI and its derivative chimeric lysine LysRODIΔAmi, lacking the amidase domain, to lyse *S. aureus* cells in pasteurized whole milk, UHT whole milk, and UHT skim milk (10^3^, 10^4^, and 10^5^ CFU/mL) was evaluated at 37 °C for 2 h. LysRODI showed a reduction between 1 and 2 log_10_ CFU/mL of *S. aureus* in the milk samples evaluated. LysRODIΔAmi was more effective in reducing *S. aureus* because there was an elimination of bacterial contamination below the detection levels in all samples, except for the UHT whole milk sample that contained 10^5^ CFU/mL of *S. aureus*, where there was a reduction of 2.82 log_10_ CFU/mL [81].

### 8.2. Endolysin-Based Control in Cheese

The biocontrol of *S. aureus* with endolysins has been little studied in cheese. However, the studies carried out have shown the potential that endolysins have to lyse the microorganism. The reduction in the bacterial cell count of *S. aureus* ATCC 33591 (10^4^ CFU/mL) during the production of ripened cheese was less than 2 log_10_ CFU/mL after 6 h of adding the endolysin Lysdb from the phage *Lactobacillus delbrueckii* at 38 °C. During cheese ripening at 10 °C after 6 weeks, *the S. aureus* counts were 2.7 × 10^3^ CFU/g [102]. On the other hand, Endolysin LysRODI (4 μM) reduced viable *S. aureus* cells during cheese production by approximately 2 log_10_ CFU/mL after 2 h incubation at 32 °C. For the sample stored for 7 days at 12 °C and 90% relative humidity, the reduction in *S. aureus* was 0.67 ± 0.44 log_10_ CFU/mL compared to the untreated control sample [81].

### 8.3. Endolysin-Based Control in Meat Products

LysSA11 endolysin at concentrations of 1.125, 2.25, 3.375, 4.5, and 9 μM showed lytic activity against MRSA CCARM 3089 (2 × 10^5^ CFU/mL) in 0.5 cm3 ham cubes at 4 °C and 25 °C for one hour. The evaluation of the lytic activity shown by LysSA11 was more efficient at 25 °C since the concentrations 2.25, 3.375, 4.5, and 9 μM completely reduced the bacteria in 30 min. In the evaluation of the inhibitory effect at 4 °C, the concentration of 2.25 μM of LysSA11 reduced viable cells to undetectable levels in 45 min [100]. The results obtained indicate that LysSA11 has greater lytic activity in the microorganism present in ham at 25 °C than at 4 °C, so treatment with endolysin can be used at 25 °C and 30 min before refrigeration of this product.

The evaluation of the lytic activity of endolysin LysSA97 (18.8 μM), carvacrol (6.66 mM) and the combination of LysSA97-carvacrol (18.8 μM and 6.66 mM) in 2 cm^3^ beef cubes contaminated with *S. aureus* (10^5^ CFU/mL) did not show a significant reduction in the live bacterial cell counts in the meat samples treated with LysSA97 and carvacrol compared to the control sample. The combination LysSA97-carvacrol reduced 2.1 ± 0.5 log_10_ CFU/cm^2^ of *S. aureus* in the meat samples for 3 h [98]. Therefore, a significant synergistic effect was presented between lysine and carvacrol, which is a monoterpenoid phenol present in the essential oils of oregano (*Origanum vulgare*), thyme (*Thymus vulgaris*), and peppermint (*Lepidium flavum*) with antimicrobial properties [103].

Biocontrol of *S. aureus* (10^4^ CFU/cm^2^) present in Chinese bacon samples (5 cm × 5 cm × 0.5 cm) containing approximately 10.52 ± 1.21% salt (180 mM) and low-salt fresh pork was carried out with LysGH15 concentrations of 0.1 nM/cm2, 0.2 nM/cm^2^, and 0.4 nM/cm^2^ at 4 °C for 2 h. The results indicated that LysGH15 increased its inhibitory effect on *S. aureus* as its concentration in Chinese bacon increased, such that the microorganism was completely reduced at the highest concentration. However, in the pork samples, there was an evident control of *S. aureus*, observing significant differences in relation to the evaluated samples, considering that with the highest concentration of LysGH15, the reduction was 1 log_10_ CFU/cm^2^ [97].

The evaluation of the inhibitory effect of the LysGH15 CHAP domain on *S. aureus* was carried out under the same conditions as previously mentioned in the same foods. However, the lytic activity of LysGH15 CHAP on *S. aureus* in Chinese bacon was lower, and the highest reduction had a value of 0.7 log_10_ CFU/cm^2^ (0.4 nM/cm^2^ of LysGH15 CHAP). In fresh pork samples, a reduction in *S. aureus* was seen as the concentration of LysGH15 CHAP increased, which completely eliminated *S. aureus* from pork. Therefore, the LysGH15 CHAP domain could be an interesting strategy for low-salt and non-acidic foods [97].

Table 2 presents the results obtained in different investigations related to the biocontrol of *S. aureus* strains and some food matrices.

## 9. Commercial Phage Products for Biocontrol of *S. aureus*

For the biocontrol of microorganisms, mixtures of bacteriophages and endolysins known as phage cocktails have been widely applied, and several commercial products are currently available for food safety applications. Among them are ListShield™ and Listex P-100™ (for *Listeria monocytogenes*), SalmoFresh™ and Salmonelex™ (*Salmonella enterica*), all approved by the United States Department of Agriculture (USDA) for use in ready-to-eat or raw foods [109]. In agriculture, the first phage-based product formally approved by U.S. regulatory agencies (U.S. Environmental Protection Agency) was Agriphage™, developed by OmniLytics Inc. in 2005 to control bacterial diseases in crops [110].

Despite the growing global interest in bacteriophages as a sustainable alternative to antibiotics, their generalized use in food for biocontrol or biopreservation remains restricted in the European Union (EU). This limitation arises because phages are not included in the list of approved food additives under current EU legislation [111]. The European Food Safety Authority (EFSA) has shown caution in endorsing phage-based applications for human consumption, mainly due to concerns regarding the potential transfer of undesirable genes and challenges in their taxonomic classifications. Consequently, bacteriophages have not been granted the Qualified Presumption of Safety (QPS) status, which has delayed regulatory acceptance compared to other regions [112].

Currently, the only phage-based preparation to obtain full regulatory approval within the EU is Bafasal^®^, authorized for use exclusively as an additive in poultry feed starting in July 2025. Meanwhile, European biotechnology companies such as Micreos have developed innovative formulations, such as PhageGuard S^TM^, designed to reduce *Salmonella* in raw and ready-to-eat meats, and PhageGuard L^TM^, targeted against *Listeria monocytogenes.* PhageGuard L^TM^ is one of the most advanced candidates under EFSA evaluation and is expected to become the first phage product authorized for direct application on food surfaces in the EU [111].

In addition to regulatory progress, several studies have demonstrated the high efficacy of commercial phage cocktails designed for *S. aureus* control. These formulations offer broader host ranges and enhanced lytic performance due to the synergistic interaction among multiple phages, which surpasses the inhibitory capacity of individual phages or endolysins [113]. For instance, the commercial product SASPject PT1.2 showed strong antibacterial activity, achieving reductions of 3–5 log_10_ CFU/mL in 93.7% of MRSA and 67.6% of MSSA isolates, as well as in mixed MRSA/MSSA cultures after only 3 h of treatment [114].

Commercial phage products specific for the control of *S. aureus* are summarized in Table 3.

## 10. Limitations of *S. aureus* Bacteriophages and Endolysins in Foods

Various researchers have studied the use of bacteriophages and endolysins for the biocontrol of *S. aureus*. Significant results have been reported in the reduction in microorganisms. Nevertheless, the application of phages and endolysins in foods presents some difficulties related to the isolation of *S. aureus* bacteriophages because they present a high level of non-specific adsorption of dead bacteria, molds, yeasts, milk proteins, carbohydrate fractions, etc., present in various samples, which generates their inactivation. Likewise, staphylococcal phages usually form small or even cloudy punctate plaques, which are difficult to enumerate or visualize, which decreases the probability of their isolation and purification [123,124,125]

Regarding the difficulty in isolating *S. aureus* bacteriophages, the isolation of phage from wastewater samples was reported after a total of 117 enrichment attempts, in contrast to the bacteriophage isolation trials for *E. coli*, *K. pneumoniae*, *P. aeruginosa*, and *Salmonella,* where there was a high success rate from the same samples [118]. In the isolation of bacteriophages of *Enterococcus faecium*, *Klebsiella pneumoniae*, *Acinetobacter baumannii*, *Pseudomonas aeruginosa*, and *Enterobacter cloacae* from 16 samples from different sources, the isolation of bacteriophages of *S. aureus* was not carried out [126].

Despite the multiple advantages presented by *S. aureus* bacteriophages and endolysins, such as their host specificity without affecting the normal microflora, self-replication in the presence of the microorganism, their use in bacterial bio-detection and as biocontrol products of *S. aureus* in foods, they have structures composed of proteins, which are affected by factors such as temperature, pH, acidity and composition of food matrices, such as fat and salt content [32,98,127]. On the other hand, horizontal gene transfer is frequently observed in *S. aureus* and is associated with the presence of prophages (group B), so the use of these bacteriophages without previously performing their genomic characterization can promote the transfer of virulent genes or resistance to antibiotics [19,36].

The implementation of bacteriophage-based products faces persistent limitations in regulation, scalability, and food matrix variability. Regulatory disparities remain a critical obstacle; although several formulations have obtained GRAS status in the U.S., the EFSA excludes phages from QPS due to classification issues, lysogenic potential, and gene transfer risks. Industrial scalability is constrained by the biological complexity of phage-host systems, complicating standardized large-scale production. Moreover, phage efficacy decreases in solid foods, where limited diffusion and adverse physicochemical conditions (pH, temperature) reduce activity. Typical bacterial reductions (1–3 log_10_ CFU/mL highlight the need for improved stabilization methods, such as encapsulation and harmonized international regulatory frameworks, to ensure broader practical adoption

In the case of endolysins, it is important to mention that their production requires the isolation, propagation, and purification of bacteriophages. In addition to the cloning of the enzymes and their purification, some endolysins show low levels of expression and insolubility [128]. In addition, the enzymatic activity of endolysins is influenced by the composition of the cell walls, so to be used in the biocontrol of Gram-negative bacteria, which have a cell wall composed of an internal cytoplasmic membrane, a peptidoglycan layer and an external membrane that has a complex lipopolysaccharide layer, the use of membrane permeabilizers such as peptides, nanoparticles, aminoglycosides, EDTA, among others, is necessary to facilitate the degradation of the cell wall of Gram-negative bacteria [129,130,131].

## 11. Conclusions

Determining the efficiency of the application of bacteriophages and endolysins for the biocontrol of *S. aureus* is of great importance in the food industry due to the advantages that phage products have compared to antibiotics and the effects on food safety associated with contamination caused by the microorganism. In the reported research, the most evaluated means for the biocontrol of *S. aureus* by bacteriophages and endolysins were in cultures of antibiotic-resistant strains and in foods such as dairy and meat products.

Bacteriophages and endolysins are made up of protein structures, so they depend on the treatment conditions for the biocontrol of *S. aureus*, such as pH, acidity, and affinity with the composition of food matrices. In addition, the lytic activity of endolysins in *S. aureus* occurs in less time compared to the use of bacteriophages and has lower bacterial resistance. It is important to note that one of the disadvantages of enzymes is that they do not have the ability to replicate as bacteriophages do.

Finally, the importance of biocontrol of *S. aureus* by bacteriophages and endolysins is highlighted, which largely inhibited the growth of the bacteria under different conditions and even in dairy and meat products, characterized by being low or high in sodium and fat.

## Figures and Tables

**Figure 1 microorganisms-13-02638-f001:**
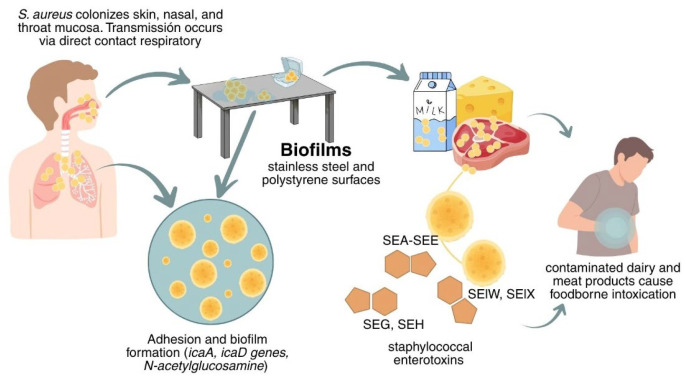
Transmission and persistence of *Staphylococcus aureus*.

**Figure 2 microorganisms-13-02638-f002:**
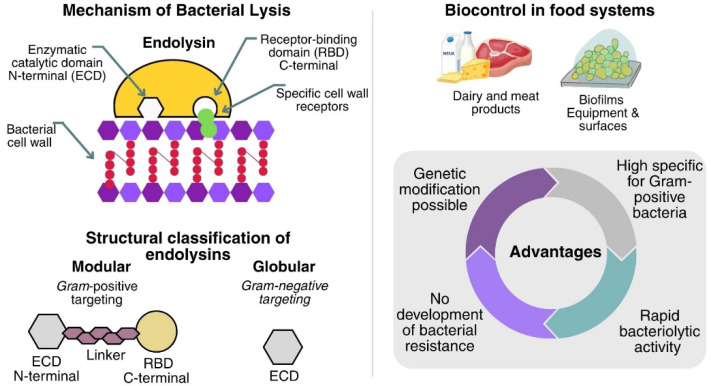
Mechanism, structural organization, and advantages of endolysins.

**Table 1 microorganisms-13-02638-t001:** *Staphylococcus aureus*-related phages.

Phage Name	ID	Old Classification/Group	Taxonomy	Genome Characteristics	Infection Cycle	Reference
UPMK_2	NC_054983.1	*Podoviridae* (A)	Virus; Duplodnaviria; Heunggongvirae; *Uroviricota*; Caudoviricetes; Azeredovirinae; Fietavirus; Fietavirus UPMK2.	40,955 bp, 62 ORFs Linear DNA	Lytic	[43].
P68	AF513033.1		Virus; Duplodnaviria; Heunggongvirae; *Uroviricota*; Caudoviricetes; Rountreeviridae; Rakietenvirinae; Rosenblum virus.	18,227 bp, 22 ORFs, Linear DNA, %GC: 29.3	Lytic	[44]
SA75	MT013111.1	*Siphoviridae* (B)	*Virus; Duplodnaviria; Heunggongvirae; Uroviricota; Caudoviricetes; Azeredovirinae; Dubowvirus*; Dubowvirus SA75.	43,134 bp, 65 ORFs, Linear DNA, %GC: 34.4	Lysogenic	[45]
SA97	NC_029010.1		*Virus; Duplodnaviria; Heunggongvirae; Uroviricota; Caudoviricetes; Azeredovirinae; Dubowvirus*; Dubowvirus SA97.	40,592 bp, 54 ORFs, Linear DNA, %GC: 34.2	Lytic	[38]
φ11	NC_004615.1		*Viruses, Duplodnaviria, Heunggongvirae, Uroviricota, Caudoviricetes, Azeredovirinae, Dubowvirus*, Dubowvirus dv11.	43,604 bp Linear DNA	Lysogenic	[46]
JD007	NC_019726.1	*Herelleviridae* (C)	*Virus; Duplodnaviria; Heunggongvirae; Uroviricota; Caudoviricetes; Twortvirinae; Kayvirus*; Kayvirus JD7.	141,836 bp, 217 ORFs, Linear DNA	Lytic	[47]
pSa-3	MF001365.1		*Virus; Duplodnaviria; Heunggongvirae; Uroviricota; Caudoviricetes; Twortvirinae; Kayvirus*.	142,827 bp, 208 ORFs, Linear DNA, %GC: 29	Lytic	[48]
Staph1N	NC_047722.1		*Virus; Duplodnaviria; Heunggongvirae; Uroviricota; Caudoviricetes; Herelleviridae; Twortvirinae; Kayvirus*; Kayvirus G1.	145,647 bp Linear DNA	Lytic	[41]

**Table 2 microorganisms-13-02638-t002:** Biocontrol studies of *S. aureus* with bacteriophages and endolysins.

Host Cell	Bacteriophage/ Endolysin	New Classification (Old Classification)	Product	Conditions and Results	References
*S. aureus* Sa9 (10^2^ CFU/mL)	Phages ΦH5 and ΦA72 (10^4^ to 10^5^ PFU/mL).	*Caudoviricetes (Siphoviridae)*	Pasteurized whole milk	The growth of *S. aureus* was inhibited regardless of whether individual phages or a combination were used; however, the mixture was significantly more effective. In pasteurized whole milk, *S. aureus* was 3.6 log_10_ lower than in the control culture without phages at the end of the incubation period at 37 °C for 8 h.	[54]
*S. aureus* Sa9 (1.7 × 10^6^ CFU/mL)	Phages phi-IPLA35 and phi-IPLA88	*Caudoviricetes (Siphoviridae)*	Fresh cheese	The phage cocktail inhibited the growth of *S. aureus* during fresh cheese production and storage at 4 °C. After 6 h from the start of treatment, the microorganism counts were below the detection limits (<10 CFU/g) and until day 14, there was no growth of *S. aureus* Sa9.	[76]
*S. aureus* Sa9 (10^4^ and 10^6^ CFU/mL)	Phages phiIPLA35 and phiIPLA88	*Caudoviricetes (Siphoviridae)*	Pasteurized whole milk	The combined treatment was able to reduce initial *S. aureus* contamination for 48 h at 25 °C, below the detection limit (<10 CFU/mL).	[104]
*S. aureus* N315	Phage SAH-1	*Caudoviricetes s (Herelleviridae)*	In vitro	At MOI 1 and 100, phage SAH-1 was able to completely lyse a culture of the strain at 38 °C–for 6 h.	[105]
*S. aureus* ATCC 25923 (10^8^ CFU/mL)	Phage pSa-3	*Caudoviricetes (Herelleviridae)*	Pasteurized whole milk	At MOIs of 0.1, 1, and 10, complete bacterial lysis occurred at 37 °C for 24 h. No viable bacterial cells were detected 24 h after inoculation (detection limit, ~10^1^ CFU/mL).	[106]
MRSA CCARM 3089 (10^5^ CFU/mL)	Endolysins LysB4EAD-LyaSA11	-	Boiled rice	Treatment of LysSA11 and LysB4EAD in combination (3.0 μM each) reduced *S. aureus* to undetectable levels within 1 h. Lytic activity was assessed at 25 °C, for 4 h.	[107]
*S. aureus* (10^7^ CFU/mL)	Endolysin LysGH15 (50 μg/mL)	-	In vitro	In the turbidity reduction test, LysGH15 was able to quickly clarify the bacterial suspension. Most of *S. aureus* was lysed into fragments.	[108]

**Table 3 microorganisms-13-02638-t003:** Characterization of phage products used for the biocontrol of *S. aureus*.

Product Name	Phages/Enzyme	New Classification (Old Classification)	Company	Presentation	Aim	Legal Approval	Reference
AB-SA01	Phages J-Sa36, Sa83, Sa87	*Caudoviricetes (Herelleviridae)*	Armata (Marina del Rey, CA, USA)	Liquid	Treatment of MRSA bacteremia	FDA	[115,116]
Enkophagum	Phages	-	Brimrose Technology Corporation (Sparks, MD, USA)	Liquid	Meat products	Commercial	[117,118]
Intestinal Bacteriophage	23 phage groups related to T4likevirus, T5likevirus, and Twortlikevirus	*Caudoviricetes (Herelleviridae* 35%), (Siphoviridae 32%), (*Podoviridae* 15%).	Eliava Biopreparations (Atlanta, GA, USA)	Liquid	Treatment for infections caused by *S. aureus*	Commercial	[119,120]
N-Rephasin^®^ SALT 200	Endolysin	-	iNtRON Biotechnology (Seongnam-si, Gyeonggi-so, South Korea)	Liquid	MRSA Treatment	Phase II	[121]
SES Bacteriophage	Phages	-	Eliava Biopreparations (Atlanta, GA, USA)	Liquid	Treatment for infections caused by *S. aureus*	Commercial	[120]
Staphefekt™	Endolysin	-	Micreos Food Safety (Wageningen, The Netherlands)	Liquid	*S. aureus,* including MRSA, on human skin	Commercial	[122]

## Data Availability

No new data were created or analyzed in this study.

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
