# Peer review of "Bacteriophages and Endolysins Used in the Biocontrol of Staphylococcus aureus"

_microorganisms, 2025, doi:10.3390/microorganisms13112638_

Round 1
Reviewer 1 Report
Comments and Suggestions for Authors
This review article provides a clear and thorough overview of bacteriophages and endolysins used to control Staphylococcus aureus in food products. The topic is timely and relevant, given the rise of antibiotic-resistant strains. The manuscript is well structured, with comprehensive coverage of phage biology, applications in dairy and meat products, and recent advances in endolysin research. The main strength of the paper is its detailed synthesis of current studies, making it a valuable reference for researchers. However, the text is sometimes dense and could be shortened for better readability. Some sections describe experimental details at length without highlighting the main conclusions. The discussion would benefit from more critical analysis, particularly on practical challenges like regulation, scalability, and variability in food conditions. Overall, the paper is well researched and informative. With minor editing for conciseness and a stronger focus on key insights, it would make a solid contribution to the field.
Author Response
|
Comments 1: However, the text is sometimes dense and could be shortened to enhance readability. Response 1: We appreciate the reviewer’s observation. In response, the text has been carefully revised to improve readability and flow by shortening dense sections and refining transitions, ensuring a clearer and more concise presentation of the main ideas. “ok” |
|
Comments 2: Some sections provide extensive descriptions of experimental details without highlighting the main conclusions. Response 2: We thank the reviewer for this valuable comment. The sections containing extensive experimental descriptions have been revised to emphasize the main findings and conclusions, improving the balance between methodological detail and critical interpretation of the results. “ok” |
|
Comments 3: The discussion would benefit from a more critical analysis, particularly concerning practical challenges such as regulation, scalability, and variability in food conditions. Response 3: We thank the reviewer for this valuable comment. In response, a detailed analysis addressing regulation, scalability, and variability in food conditions has been included in Section 10, Limitations of S. aureus bacteriophages and endolysins in foods. The implementation of bacteriophage-based product face persistent limitations in regulation, scalability, and food matrix variability. Regulatory disparities remain a critical obstacle; although several formulations have obtained GRAS status in the U.S., the EFSA excludes phages from QPS due to classification issues, lysogenic potential, and gene transfer risks [111]. Industrial scalability is constrained by the biological complexity of phage-host systems, complicating standardized large-scale production [128]. Moreover, phage efficacy decreases in solid foods, where limited diffusion and adverse physico-chemical conditions (pH, temperature) reduce activity. Typical bacterial reductions (1 – 3 log10 CFU/Ml highlight the need for improved stabilization methods such as encapsula-tion and harmonized international regulatory frameworks to ensure broader practical adoption [129].
Comments 4: Overall, the article is well-documented and informative. With minor editing for greater conciseness and a sharper focus on key ideas, it would constitute a strong contribution to the field. Response 4: We sincerely thank the reviewer for the positive and encouraging assessment. Minor editorial adjustments were made to enhance the manuscript’s conciseness and emphasize the key ideas, ensuring greater clarity and a stronger overall contribution to the field.
|

Reviewer 2 Report
Comments and Suggestions for Authors
Thank you for the opportunity to review this paper. The paper is well-written and the topic is sound. Although the used literature is appropriate, I recommend to integrate some novel references on engineered phages and synthetic endolysins (from past few years). Also, there are some suggestions to improve the paper quality:
- I suggest that authors rephrase the abstracts part. It repeats general food safety information. Try to make it attractive and emphasize novel aspects.
- Add MRSA in the keywords
- Lines 117-125: The explanation is missing molecular regulation detail
- Lines 167-170: The authors should clarify genome compaction statement. Try to mention how this limits horizontal gene transfer and increases biosafety suitability.
- In section 4, the authors should add a discussion on phage stability and environmental persistence in food.
- In section 5.1. the authors missed to explain how phage cocktail optimization is driven?
- Lines 457-467: the part about cheese is too short. Try to expand it with encapsulation and immobilization mechanisms (chitosan, polymer coatings)
- In Section 9. What is the current status of EU-authorized products such as PhageGuard S.
Author Response
|
Comments 1: I suggest that the authors reformulate the abstract. It repeats general information about food safety. Try to make it more engaging and highlight the novel aspects. Response 1: The abstract has been reformulated to better emphasize the relevance of S. aureus biocontrol in the food industry using bacteriophages. General information on food safety was reduced, and the novel aspects and focus of the review are now highlighted more clearly. “Staphylococcus aureus is a major foodborne pathogen associated with contamination of dairy and meat products, posing a persistent challenge to food safety due to its biofilm formation and resistance to multiple antibiotics. In this review, we summarize recent advances in the use of bacteriophages and phage-derived endolysins as targeted biocontrol agents against S. aureus in food systems. Bacteriophages exhibit host specificity and self-replicating capacity, while endolysins provide rapid lytic activity, minimal resistance development, and effectiveness against biofilm-embedded cells. Studies demonstrate significant microbial reductions in milk, cheese, and meat matrices, although factors such as pH, salt, and fat content can influence their efficacy. The integration of these biocontrol tools into food preservation represents a sustainable and safe alternative to conventional antimicrobials. Finally, we discuss current limitations and the need for optimizing formulations, stability, and regulatory frameworks to facilitate the adoption of phage and endolysin-based products in the food industry.”
Comments 2: Add MRSA to the keywords. Response 1: Thank you for pointing this out. We agree with this comment. Keywords: antimicrobial strategies; bacteriophages; endolysins; food security; food in-dustry, Methicillin-resistant Staphylococcus aureus (MRSA).
Comments 3: Details about molecular regulation are missing from the explanation. Response 1: A more detailed description of the molecular regulation involved in S. aureus adhesion and biofilm formation has been added, as shown in the following text. Biofilm development in S. aureus is largely regulated by the intracellular adhesion (ica) locus, which encodes enzymes involved in the synthesis of the polysaccharide intercellu-lar adhesin (PIA). The icaA and icaD genes play a central role in the process, catalyzing the polymerization of N-acetylglucosamine, the main exopolysaccharide that mediates intracellular adhesion and biofilm accumulation. The expression of the ica operon is fur-ther modulated by global regulators such as sarA, which enhances biofilm formation, and agr, which typically represses it in favor of acute virulence. Nevertheless, biofilm mass can vary even among strains harboring the ica locus, suggesting the involvement of ica-independent mechanisms, including surface proteins such as protein A and fibron-ectin-binding proteins [29]. Strains of S. aureus isolated from foods, processing facilities, and food handlers have demonstrated the ability to form biofilms on both stainless steel and polystyrene surfaces [30].
Comments 4: Lines 167–170: The authors should clarify the statement regarding genome compaction. They should explain how this limits horizontal gene transfer and enhances biosafety suitability. Response 1: A more detailed explanation of the genome compaction of S. aureus phages has been provided, as shown in the following text. These phages exhibit remarkable genome compaction (<20 kb) and a streamlined genetic organization primarily dedicated to replication and structural functions, leaving minimal space for accessory or virulence-related genes. Their compact and well-defined genomes, together with a strictly lytic lifestyle, restrict uncharacterized regions and mobile genetic elements, thereby reducing the potential for horizontal gene transfer and enhancing their biosafety for therapeutic and biocontrol applications [39].
Comments 5: In Section 4, the authors should include a discussion about phage stability and their environmental persistence in foods. Response 1: We agreed to expand the section by including a discussion on the stability of bacteriophages under different environmental conditions, as well as the environmental considerations related to their use in food systems Beyond lytic efficiency, the stability of S. aureus bacteriophage under varying physicochemical and environmental conditions is a key determinant of their applicability in food systems. Phages generally remain viable between 4 °C and 50 °C and within a pH range of 4.0 – 11.0, but they are rapidly inactivated at temperatures ≥ 70 °C or under extreme acidic (pH ≤ 3.0) environments [53]. In complex food denature phage particles, diminishing infectivity [54]. To enhance stability, preservation strategies such as lyophilization with crioprotectans (e.g., trehalose, sucrose) or microencapsulation within polymeric matrices (e.g., alginate, chitosan) have been successfully applied to maintain structural integrity and lytic functionality under adverse condition [55]. Importantly, the environmental impact of phage-based interventions is considered minimal, as bacteriophages specifically infect bacterial hosts and do not affect eukaryotic cells. Their high specificity and natural biodegradability position them as sustainable alternatives to chemical disinfectants or antibiotics, minimizing non-target effects and eco-logical disturbances. To ensure biosafety, only strict lytic phages should be employed thereby avoiding risks of lysogeny and horizontal gene transfer [56]. Supporting this, in situ assessments demonstrated that phages such as FoX2 and FoX4 did not alter local biomass or microbial diversity in soil microbiomes, confirming their environmental com-patibility and alignment with sustainable biotechnological [57].
Comments 6: In Section 5.1, the authors omitted an explanation of how phage cocktail optimization is driven. Response 1: Thank you for pointing this out. We have expanded the information regarding the optimization of phage cocktails. Similarly, when evaluated in raw milk under the same conditions, the maximum reduc-tion achieved was around 1 log10 CFU/mL [68]. The use of bacteriophage cocktails offers an effective alternative for controlling foodborne pathogens such as S. aureus. Their com-bined action broadens the host ranfe and decreases the likelihood of bacterial resistance. Optimizing their formulation involves selecting complementary phages that maximize bacterial reduction while maintaining manageable complexity. This strategy enhances robustness and ensures sustained lytic efficacy across diverse strains and food matrices [69].
Comments 7: Lines 457–467: The section on cheese is too brief. Try expanding it to include encapsulation and immobilization mechanisms (chitosan, polymeric coatings). Response 1: We have expanded Section 5.2 “Cheese” with additional studies on the application of bacteriophages. The phage cocktail (vB_SauS-phi-IPLA35 and vB_SauS-phi-IPLA88) effectively re-duced S. aureus in cheese. In fresh cheese, counts decreased by 3.83 log CFU/g within 3 h, reaching undetectable levels at 6 h. In hard cheese, reductions reached 4.64 log CFU/g in the curd and 1.24 log CFU/g after ripening, demonstrating consistent efficacy during pro-cessing and storage [76]. Similarly, the S. aureus phage cocktail (EBTH and K2, 108 PFU/mL) efficiently controlled both inoculated and native S. aureus in traditional Egyptian chesses. In soft varieties (Karish and Domiati), bacterial counts dropped below detection (≤ 10 CFU/g) within 3 – 24 h, preventing enterotoxin accumulation, whereas in hard Ras cheese, complete elimination occurred after 60 – 90 days of ripening [77]. More recently. a biopreservation strategy for hard Dutch-type cheese using two virulent phages (No. 4 and No. 8) applied post-pasteurization (108 PFU/mL) completely prevented S. aureus growth during 60 days of ripening, while the control reached 154.2 CFU/g, confirming the strong protective capacity of the phage treatment [78]. Collectively, these studies demonstrate that the use of lytic bacteriophage and phage cocktails can effectively reduce or eliminate S. aureus in both fresh and hard cheeses with-out compromising the activity of starter cultures or altering the physicochemical proper-ties of the final product. This highlights their potential as safe and efficient biocontrol agents in the dairy industry.
Comments 8: In Section 9, what is the current status of EU-authorized products such as PhageGuard S.? Response 1: Additional information has been included regarding the current regulatory status of phage-based products in the European Union. Despite the growing global interest in bacteriophages as a sustainable alternative to antibiotics, their generalized use in food for biocontrol or biopreservation remains re-stricted in the European Union (EU). This limitation arises because phages are not in-cluded in the list of approved food additives under current EU legislation [111]. The Eu-ropean Food Safety Authority (EFSA) has shown caution in endorsing phage-based ap-plications for human consumption, mainly due to concerns regarding the potential trans-fer of undesirable genes and challenges in their taxonomic classifications. Consequently, bacteriophages have not been granted the Qualified Presumption of Safety (QPS) status, which has delayed regulatory acceptance compared to other regions [112]. Currently, the only phage-based preparation to obtain full regulatory approval within the EU Bafasal®, authorized for use exclusively as an additive in poultry feed starting in July 2025. Meanwhile, European biotechnology companies such as Micreos have devel-oped innovative formulations PhageGuard STM, designed to reduce Salmonella in raw and ready-to-eat meats, and PhageGuard LTM, targeted against Listeria monocytogenes. Phage-Guard LTM, is one of the most advanced candidates under EFSA evaluation and is expected to become the first phage product authorized for direct application on food surface in the EU [111]. In addition to regulatory progress, several studies have demonstrated the high effi-cacy of commercial phage cocktails designed for S. aureus control. These formulations offer broader host ranges and enhanced lytic performance due to the synergistic interaction among multiple phages, which surpasses the inhibitory capacity of individual phages or endolysins [113]. For instance, the commercial product SASPject PT1.2 shown strong an-tibacterial activity, achieving reductions of 3 – 5 log10 CFU/mL in 93.7% of MRSA and 67.6% of MSSA isolates, as well as in mixed MRSA/MSSA cultures after only 3 h of treat-ment [114]. |

Round 2
Reviewer 2 Report
Comments and Suggestions for Authors
The authors have addressed all my suggestions and improve overall quality of the paper. Therefore, I recommend it for the publication in the present form.